# Research Trends in the Use of Machine Learning Applied in Mobile Networks: A Bibliometric Approach and Research Agenda

Vanessa García-Pineda [1], Alejandro Valencia-Arias [2,*], Juan Camilo Patiño-Vanegas [3], Juan José Flores Cueto [4], Diana Arango-Botero [3], Angel Marcelo Rojas Coronel [5] and Paula Andrea Rodríguez-Correa [6]

1   Facultad de Ingeniería, Corporación Universitaria Americana, Medellin 055428, Colombia; vgarcia@americana.edu.co
2   Escuela de Ingeniería Industrial, Universidad Señor de Sipán, Chiclayo 14001, Peru
3   Facultad de Ciencias Económicas y Administrativas, Instituto Tecnológico Metropolitano, Medellin 050034, Colombia; juanpatino@itm.edu.co (J.C.P.-V.); dianaarangob@itm.edu.co (D.A.-B.)
4   Unidad de Virtualización Académica, Universidad de San Martin de Porres, Santa Anita 15011, Peru; jfloresc@usmp.pe
5   Escuela de Ingeniería Mecánica, Universidad Señor de Sipán, Chiclayo 14001, Peru; rmarcelo@crece.uss.edu.pe
6   Centro de Investigaciones, Institución Universitaria Escolme, Medellin 050012, Colombia; cies4@escolme.edu.co
*   Correspondence: valenciajho@crece.uss.edu.pe; Tel.: +57-3002567977

**Abstract:** This article aims to examine the research trends in the development of mobile networks from machine learning. The methodological approach starts from an analysis of 260 academic documents selected from the Scopus and Web of Science databases and is based on the parameters of the Preferred Reporting Items for Systematic Reviews and Meta-Analyses (PRISMA) statement. Quantity, quality and structure indicators are calculated in order to contextualize the documents' thematic evolution. The results reveal that, in relation to the publications by country, the United States and China, who are competing for fifth generation (5G) network coverage and are responsible for manufacturing devices for mobile networks, stand out. Most of the research on the subject focuses on the optimization of resources and traffic to guarantee the best management and availability of a network due to the high demand for resources and greater amount of traffic generated by the many Internet of Things (IoT) devices that are being developed for the market. It is concluded that thematic trends focus on generating algorithms for recognizing and learning the data in the network and on trained models that draw from the available data to improve the experience of connecting to mobile networks.

**Keywords:** mobile networks; mobile communication systems; 5G; PRISMA; machine learning; internet of things

## 1. Introduction

The availability of the current network has become an aspect of interest and a central concern for all individuals and organizations [1]. Mobile networks allow people to be constantly connected, regardless of where and when they are, especially if they are outdoors. Despite the progress that currently exists in regard to hardware and software development, there are still shortcomings in the availability of mobile connectivity services [2]. Cellular telephone operators do not yet have complete network expansion, and the antennas used in some places do not achieve enough range for total signal coverage [3]. In addition, future communication networks must address the scarce spectrum in order to adapt to the great growth in heterogeneous wireless devices. In this sense, efforts are being made to address the coexistence of the spectrum, improve knowledge concerning it, reinforce

scheme authentication to improve spectrum monitoring and spectrum management and enable secure communications, among other things [4].

However, software-defined networks (SDNs), network function virtualization (NFV) and cloud computing are receiving significant attention in fifth generation (5G) networks [5]. With the advancement of different optimization techniques, methods and tools (such as artificial intelligence, machine learning and big data), it is possible to improve the availability, quality and coverage of mobile networks, thereby facilitating better service provision to users of mobile networks [2]. Future smart wireless networks require an adaptive learning approach towards a shared learning model to enable collaboration between data generated by network elements and virtualized functions [6].

Additionally, 5G communication networks aim to provide a paradigm shift of the wireless spectrum at higher frequencies to meet the demands of large traffic volumes, extreme transmission speed, low traffic latency and massive connectivity; this paradigm shift places human society in a new service model that is prepared for the demands of the Internet of Things (IoT) and provides a better model of data processing through the use of edge servers to improve the protection of data privacy [7].

Given the advances and great demand in connectivity that society currently demands due to different factors, such as the IoT, smart cities and the growth in wearable devices (wearables), the better operation and management of networks through algorithms that learn and use available data and measurements to optimize network performance is needed [3]. This requirement can be achieved by implementing technologies such as AI, machine learning and big data; however, applying this type of approach to planning, designing, managing and operating networks is still in its early stages, because existing network architectures do not adapt to the networks enabled for these technologies [8].

Based on the above, the main objective of this article is to examine the machine learning-based research trends regarding the development of mobile networks by performing a bibliometric analysis of the Scopus database. A main conclusion is that the main trends in the use of Industry 4.0 technologies, such as machine learning, involve designing and developing algorithms that can learn from the data provided by network traffic and that allow the current and next-generation networks to self-manage in such a way that they respond to the needs and demands of users and the different types of devices connected to the network.

In addition, with the purpose of guiding the development of the research and achieving the stated objective, this study also outlines the following research questions:

RQ1: In which years was there the greatest interest in mobile networks and machine learning?

RQ2: What are the main research references in the scientific literature?

RQ3: How has the literature on mobile networks and machine learning conceptually evolved?

RQ4: What are the main growing and emerging themes derived from the scientific production on mobile networks and machine learning?

RQ5: What elements should a researcher include in their future work on mobile networks and machine learning?

In this regard, this research is composed of the introductory section above, which explains in detail the conceptualization and importance of the topic, as well as the objective and the respective research questions. Then, the Materials and Methods section will detail the necessary procedures to achieve the results that support the investigation. Subsequently, the bibliometric results of the research are developed, as well as a thematic discussion and research trends. The final section outlines the conclusions reached in this article.

For the reader's ease, the definitions of the abbreviations used in the text are presented in Table 1.

**Table 1.** Abbreviations.

| Abbreviations | Meaning |
|---|---|
| 3G | Third generation |
| 5G | Fifth generation |
| 6G | Sixth generation |
| AI | Artificial intelligence |
| ANNs | Artificial neural networks |
| UAVs | Aerial vehicles |
| B5G | Beyond 5G |
| DRL | Deep reinforcement learning |
| DL | Deep learning |
| FFNN | Feedforward neural network |
| FL | Federated learning |
| HetNets | Heterogeneous networks |
| KPIs | Key performance indicators |
| LSTM | Long short-term memory |
| IoT | Internet of Things |
| ML | Machine learning |
| mmWave | Millimeter wave communications |
| MIMO | Multiple inputs, multiple outputs |
| ML | Machine learning |
| NFV | Network function virtualization |
| PRISMA | Preferred Reporting Items for Systematic Reviews and Meta-Analyses |
| QC | Quantum computing |
| QML | Quantum ML |
| RAN | Radio access networks |
| RIS | Reconfigurable intelligent surfaces |
| RNNs | Recurrent neural networks |
| SDNs | Software-defined networks |
| THz | Terahertz |

## 2. Materials and Methods

In accordance with the objective of the research, a methodological design is proposed, which, based on the minimum parameters established by the Preferred Reporting Items for Systematic Reviews and Meta-Analyses (PRISMA) statement for literature reviews, enables a bibliometric analysis that allows, in accordance with [9], the mapping of the scientific literary body currently available in the databases. This is carried out in order to identify, on the one hand, the current research trends around the proposed topic and, on the other hand, the future directions that can be seen, in both a statistical analysis and positional keywords, as the main metadata topics of scientific activity. In this sense, regarding the execution of the methodological design, the studies by [10,11] are followed, whereby the items or detailed parameters established by the PRISMA declaration are related and specified in the following subsections.

### 2.1. Eligibility Criteria

The eligibility criteria, as evidenced in [10], are those that enable the specification and the detailing of the characteristics which the studies analyzed in the literature review process must meet. Therefore, for the methodological design used in this study, the following inclusion and exclusion criteria were established.

2.1.1. Inclusion Criteria

As the stipulated in the inclusion criteria for this literature review, the research to be analyzed must involve, with their titles and keywords as the primary bibliographic metadata, the different combinations by which mobile connection networks are known, in combination with the subject of machine learning. This inclusive combination refers to the Boolean operator "AND".



Likewise, to gain a further understanding the scope of the review, inclusion criteria are established that account for aspects such as the type of publication, the publication status and metadata registration, as well as the rigor of each of these, specifically with regard to the inclusion of journal articles or book chapters that contain complete metadata for a holistic analysis of the information.

### 2.1.2. Exclusion Criteria

Therefore, in a complementary way, there are exclusion criteria that, in accordance with the reference items of the PRISMA statement, involve different phases. The first is known as screening, whereby all bibliographic records that do not correspond to the format stipulated by the inclusion criteria are excluded. Conference proceedings are generally excluded. Likewise, research that contains incomplete metadata are eliminated, as are all other papers that do not contain defined evaluation criteria, so that the results can guarantee scientific rigor. Then, there is a second exclusion phase, called eligibility. Although they were not excluded in the first phase, all papers that do not involve relevant elements of analysis are eliminated or excluded based on the thematic approach raised in the research.

### 2.2. Information Sources

After the eligibility criteria of the literature review are defined, in accordance with the protocol established by the PRISMA guidelines, a phase occurs during which one defines the sources of information for the methodological design. In this sense, with the understanding that all literature reviews are based on secondary sources of information and that the scope of this research consists of a review of scientific literature, the two current main academic and scientific databases, namely, the Web of Science and Scopus, are established as sources of information, as they are complete, robust, detailed and rigorous suppliers of metadata, involving publications and scientific activity from institutions, which are found in the bibliography, aided by the interface developed by the databases, thus improving the performance of the review processes [12].

However, as demonstrated in [10], the phase in which the sources of information are established is contained by the time period that is used in the search for scientific information. Thus, it must be mentioned that the present review process dates from the first articles that are available within each database up to the present time; therefore, while a defined interval is not evident, the evolutionary process that the literary and scientific body of international research undergoes regarding use of machine learning for connecting to mobile networks can be understood.

### 2.3. Search Strategy

To improve the levels of rigor, detail and replicability of the methodological design, the PRISMA guidelines suggest the search strategy that will be executed for each of the information sources selected for the research. Therefore, regarding the details of this section, it is essential to mention that the strategy responds to two interconnected elements: the interface of the information source and the inclusion criteria described. In this sense, it is understood that since the relevant databases are international, their interface requires the realization of a metadata search in English. For this, the following specialized search equations are designed that include all the inclusion criteria described as well as the inherent search characteristics of the database:

Web of Science Search:

((TI = ({Mobile Telecommunication System} OR {Mobile Network} OR {5G Mobile Communication System} OR {Cellular Network} OR {5G mobile} OR {6G mobile}) AND TI = ({Machine learning}))) OR ((AK = ({Mobile Telecommunication System} OR {Mobile Network} OR {5G Mobile Communication System} OR {Cellular Network} OR {5G mobile} OR {6G mobile}) AND AK = ({Machine learning})))

Scopus Search:

((TITLE ({Mobile Telecommunication System} OR {Mobile Network} OR {5G Mobile Communication System} OR {Cellular Network} OR {5G mobile} OR {6G mobile}) AND TITLE ({Machine learning}))) OR ((KEY ({Mobile Telecommunication System} OR {Mobile Network} OR {5G Mobile Communication System} OR {Cellular Network} OR {5G mobile} OR {6G mobile}) AND KEY ({Machine learning})))

*2.4. Study Record*

After the specific search strategy was applied within each of the databases selected as sources of information, 1871 documents published during the time interval of 2006 to 2022 were obtained. Of these documents, 1468 were from the Scopus database, and the remaining 403 were from the Web of Science database. However, according to the PRISMA referencing guidelines, Refs. [10,11] suggest providing a detailed outline of our use of these documents. Thus, we provide the following information.

2.4.1. Data Management

In order to manage the bibliographic data obtained from the selected information sources, Microsoft Excel® Office tools and the free access software VOSviewer version 1.6.19, through which all the metadata obtained were stored and processed, were utilized. This data treatment necessarily alludes to information homogenization, whereby the information derived from both databases was standardized so that both types of information had the same format; this standardization started with the elimination of duplicate records and the consolidation of a unified database.

2.4.2. Selection Process

Once the information was stored in the aforementioned tools and the process of eliminating duplicate records was carried out, a unified database was designed, which was provided to two of the authors of this research, who independently reviewed the application of the exclusion criteria established at the beginning of the methodological design. This independent analysis guarantees a reduction in the bias that can be derived from the selective analysis of information [10,11]. Finally, to increase the level of detail and replicability of the methodological design, we provide Figure 1, which summarizes all the steps described by the PRISMA statement for literature reviews.

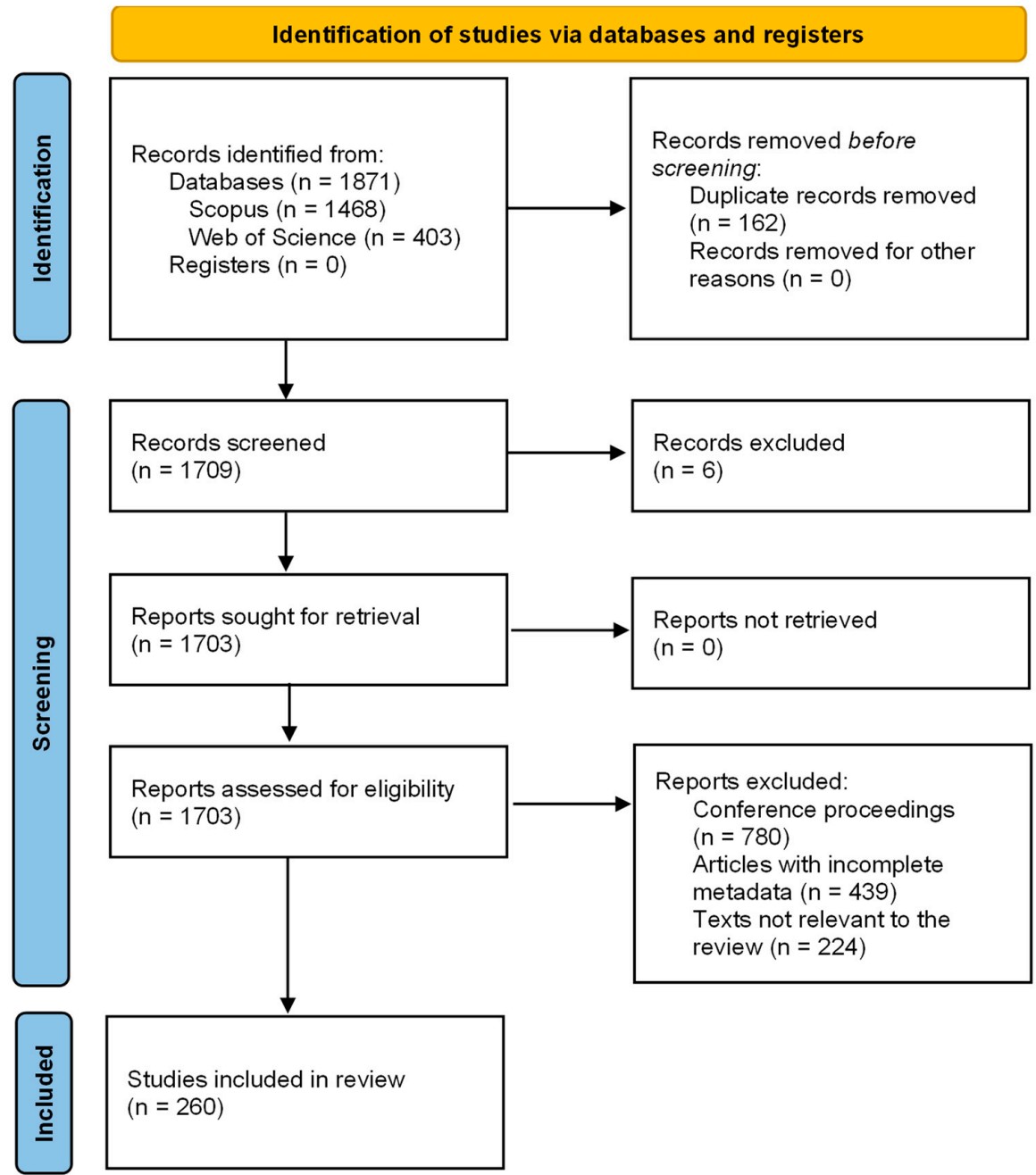

**Figure 1.** PRISMA 2020 flow diagram. Our own depiction.

### 3. Results

Based on the results obtained, the data from the different publications were analyzed, starting with the number of publications per year. Figure 2 shows how the number of publications began to increase in 2017, maintaining this trend in 2018, before experiencing a significant increase in both 2019 and 2020. In 2019, among the published articles with the most citations, the work of [13], with 664 citations, is the most cited paper on the subject of mobile networks and machine learning. In their work, the authors perform a survey in which they examine deep learning and wireless mobile networks in order to study the degree of acceptance of said technologies. With 315 citations, a proposal of an antenna arrangement modulating frequencies by applying MIMO (multiple inputs, multiple outputs) for mobile networks [14] had the next highest number of citations.

## Publications per year

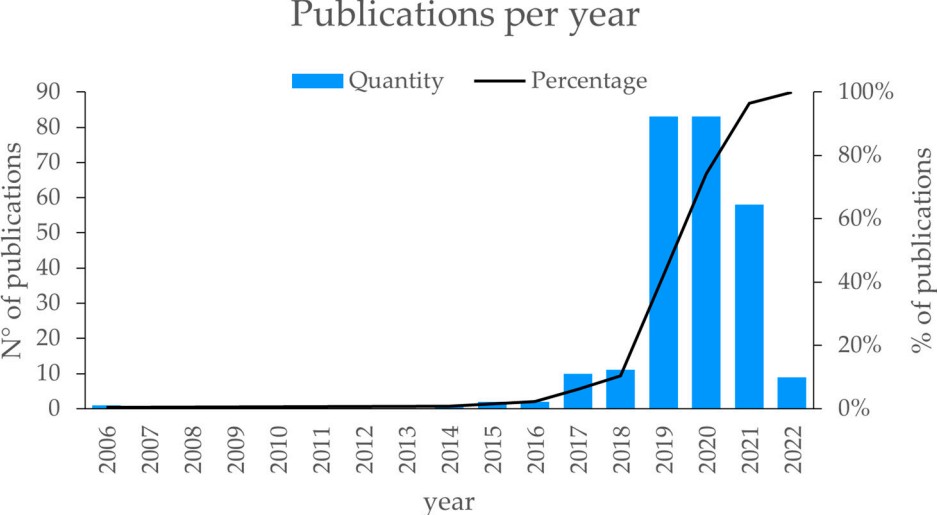

**Figure 2.** Number of publications per year concerning mobile networks and machine learning. Own graph.

In 2020, the publication with the highest number of 272 citations was a study that carried out a survey on multi-access edge computing in 5G, the technology used, the fundamentals and basic concepts and the integration of different technologies [15]. The research with the second-highest number of citations, at 160 citations, comprised study that analyzed the challenges that the industry faces regarding the advancement of technologies based on deep learning for 6G networks [16]. In 2021, an article [16] with 206 citations was published, which similarly analyzed the path toward sixth generation (6G) networks in the future [17]. In the year 2022, one of the most cited articles, with 37 citations, focused on the development of a scheme to hide reversible data in 5G systems by using deep learning techniques [18].

Regarding the main journals, in Figure 3, the IEEE Access journal is shown to have the most publications, accounting for 38% of the total publications on the subject of machine learning and mobile networks, with 98 publications on the subject. In addition, it is positioned as the journal with the highest scientific impact in terms of citations, with more than 2190 citations of its scientific publications. Its most cited publications are [13,19], with 650 citations, and the research carried out was a survey in which the applications of deep reinforcement learning for mobile communication networks was analyzed.

In addition, another of the main journals in the field of research is IEEE Communications Surveys and Tutorials, which, although it is not positioned as a leading journal in terms of scientific productivity, since it has only 11 publications associated with the subject, these are highly relevant to the field, judging from the total number of citations. Among the main contributions of this journal to the scientific body, is a study that delves into the existing knowledge concerning the relationship between deep learning and mobile wireless networks, concluding with the best adoption techniques [13].

Regarding the most highly contributing countries on the subject of machine learning in mobile networks, Figure 4 shows that the United States has the highest number of publications, with 60 publications and more than 2300 citations, making it the main country for scientific publications on the subject. Among the most relevant articles that have been published in the United States are [14,20]. The latter examined the internet of industrial using a cyber-physical systems approach. The next most prolific country is the United Kingdom, with 1572 citations in 38 publications, including two articles that discuss deep learning in mobile and wireless networks and carry out analyses via surveys [13,15], both have the most citations from among articles from the United Kingdom.

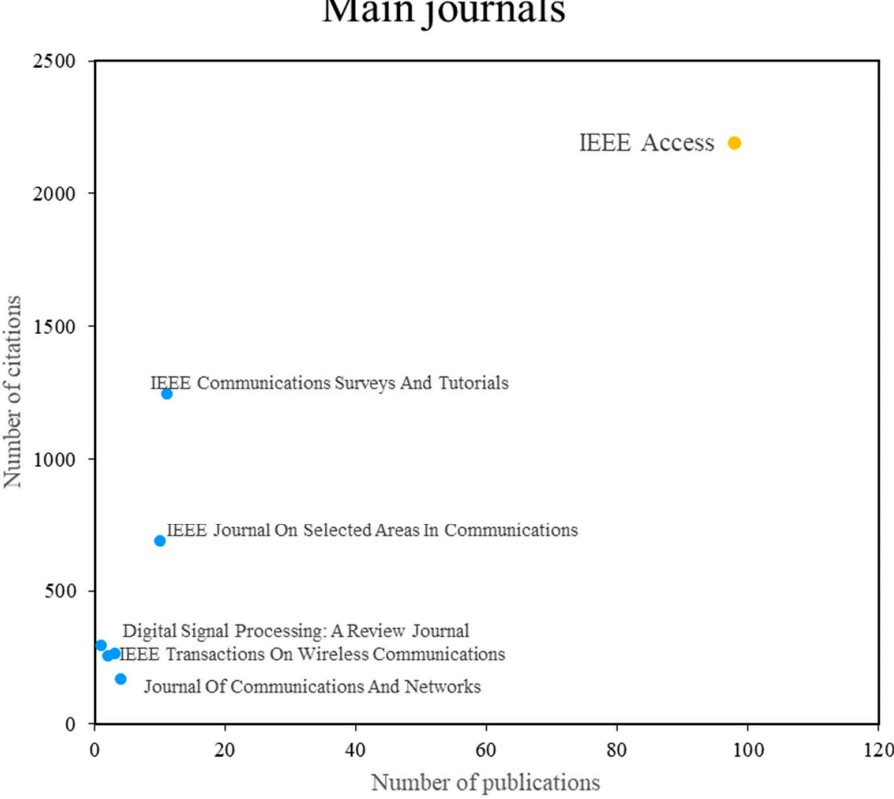

**Figure 3.** Main journals on mobile networks and machine learning. Own illustration.

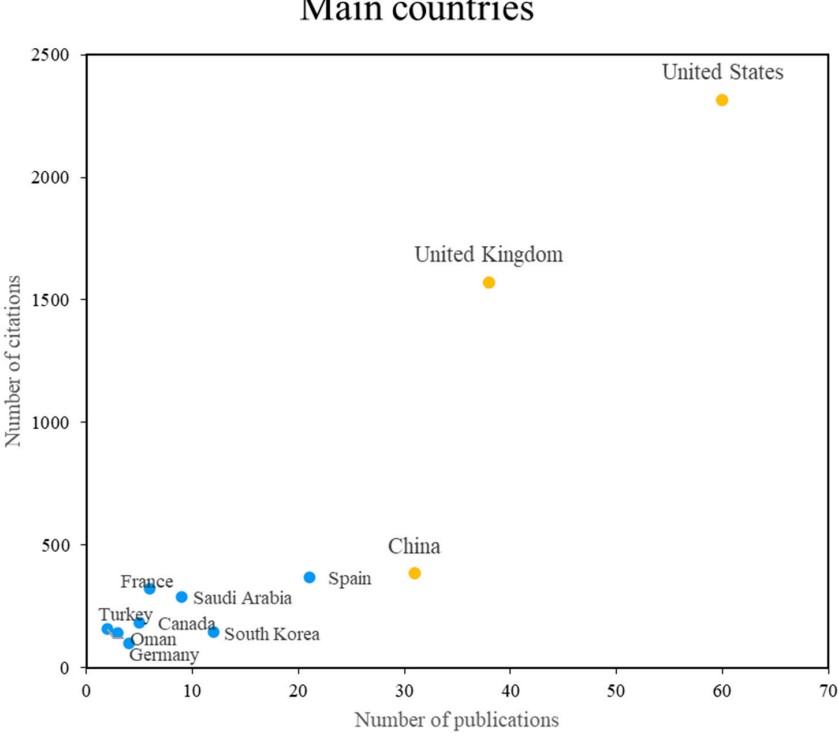

**Figure 4.** Main countries contributing to research on mobile networks and machine learning. Own graph.

In third place is China, with 31 publications. One of these publications has the most citations of any publication thus far with 650 citations. This work comprises the analysis of various applications of deep reinforcement learning in communication systems and networks [19]. Another work, which has 180 citations thus far, examines the use of blockchain and deep reinforcement learning to enhance 5G smart grids [21]. Spain is the next country with the most publications, 21 in total, with more than 360 citations, of which the work with the most citations is a review of the transition from 4G to 5G and the application of machine learning in mobile networks [22].

In addition, the present bibliometric analysis examines the institutions that have led scientific research regarding the use of machine learning in mobile networks, as shown in Figure 5, evaluating scientific productivity as well as academic impact. Here, it can be observed that the University of Oulu stands out, ranking as the institution that publishes the most in this field, as well as being the institution with the third most cited publications.

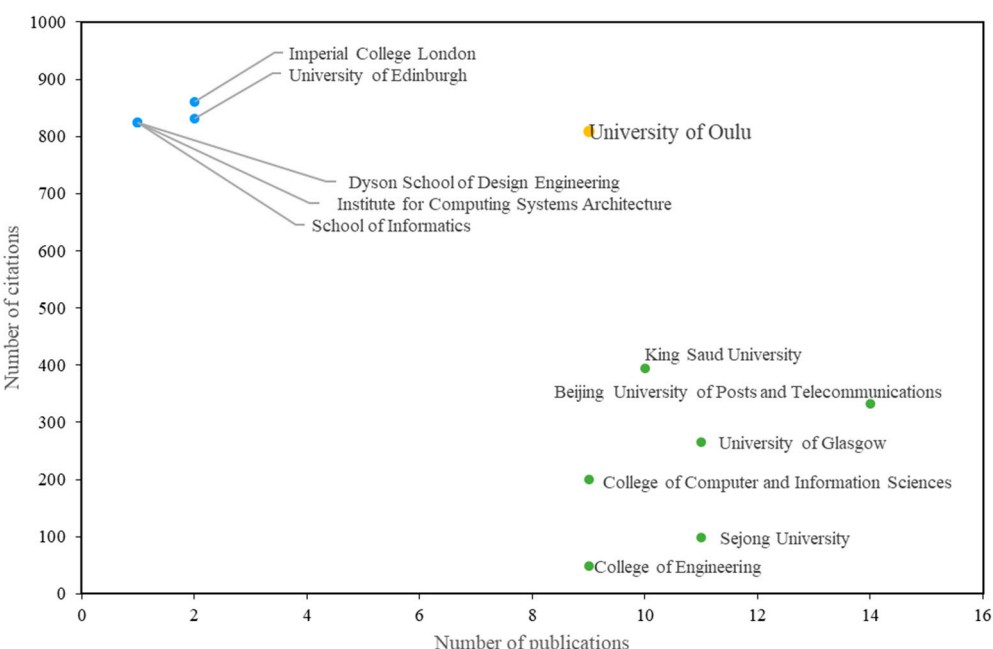

**Figure 5.** Main institutions publishing on mobile networks and machine learning. Own graph.

Important research studies have emerged from this institution, exploring the foundations of three-dimensional wireless cellular networks in unmanned aerial vehicles [23], addressing the management of large volumes of data in the telecommunications context, proposing perspectives of efficiency and performance in mobile networks [24], as well as other studies that have focused on the characteristics of THz wireless systems, which are essential for the development and understanding of wireless systems [25].

Next is the Imperial College London, which, although not prominent in terms of scientific productivity, has had a high impact through the number of citations, making its contributions relevant to the scientific literature on the subject. This institution has delved into aspects such as the use of deep learning in wireless networks [13], as well as the use of deep learning for the management and orchestration of virtual network functions [13].

Likewise, the Beijing University of Posts and Telecommunications stands out among the main institutions publishing scientific literature [26]. However, it stands out due to its significant scientific productivity, which demonstrates its commitment to knowledge generation, even though it is currently not among the institutions with the highest impact in terms of citations.

This institution has produced different research approaches, ranging from the study of algorithms based on a scalable Gaussian process for wireless traffic prediction [27],

the evolution of non-orthogonal multiple access techniques that enable the discussion of 6G technology [28], as well as approaches based on machine learning for the flexible programming of transmission time intervals in coexistence with eMBB and uRLLC as one of the main challenges in managing mobile network resources [29].

Figure 6 presents the main authors publishing on this subject. The most cited researcher on the subject is Debbah, M., with 674 citations in total; this author is also the most scientifically productive, with eight publications concerning machine learning and mobile networks. His most cited work, with 239 citations in total, is "Wireless Network Intelligence at the Edge", in which the authors explored the main components of edge machine learning to propose different divisions of neural networks for intelligent wireless networks [30]; additionally, a co-author of this article is one of the other most prolific authors on this subject, namely Bennis, M., with six publications and 650 citations in total. Then, Haddadi, H.; Patras, P.; and Zhang, C., are the next most cited authors, with 664 citations on their work entitled "Deep Learning in Mobile and Wireless Networking: A Survey" [13]. Chan, K.; He, T.; Leung, K.K.; Makaya, C; and Salonidis, T., also research networking and are the next most cited authors with 518 citations; their most cited work is "Enabling Massive IoT Towards 6G: A Comprehensive Survey" [31].

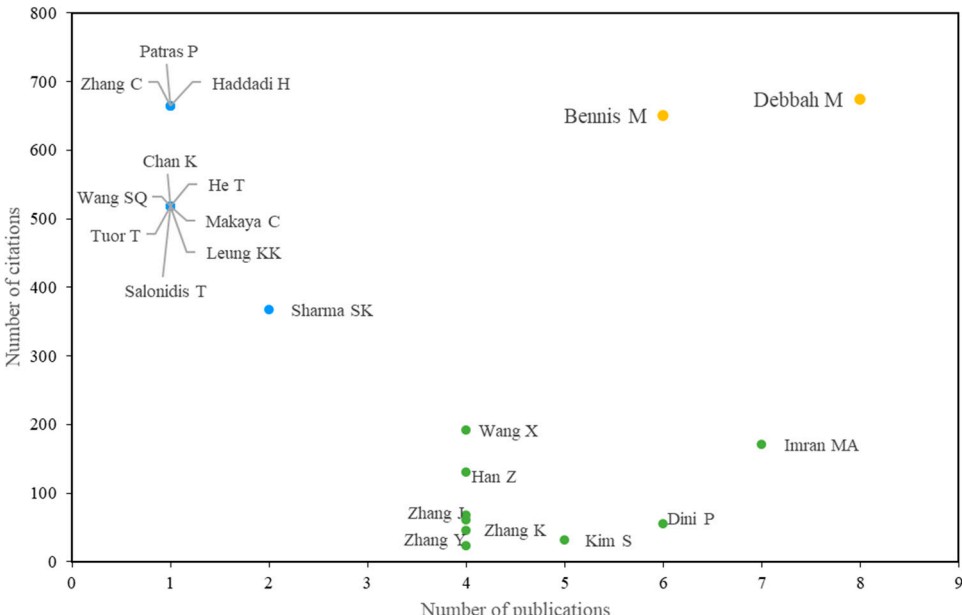

**Figure 6.** Most prolific authors publishing on the subject of mobile networks and machine learning. Own graph.

Regarding keyword patterns, Figure 7 shows the validity and frequency of the keywords according to their recurrence by year. The most frequent and current terms are shown in Quadrant 1 of Figure 7. Three terms are associated with the advancement of 5G and 6G networks because the accelerated growth of mobile networks has allowed the generation of new versions and modifications for the connectivity of mobile devices in cellular networks, in addition to advances that increase both the supply and demand in connectivity resources [32]. Research on 5G networks has focused on optimizing the resources of this network, mainly to guarantee availability and quality in the provision of services for IoT devices [33] in addition to the correct addressing for transmitting and receiving a signal. In these studies focused on the improvement of 5G networks, the terms "artificial intelligence", "Internet of Things" and "deep learning" were used, which are also the next three most common terms in this quadrant.

**Figure 7.** Validity quadrants in the area of mobile networks and machine learning. Own graph.

For artificial intelligence, the different techniques of this technology are focused on improving the management of mobile networks by defining characteristics for 5G network software with efficient, agile and mainly autonomous and cognitive management [34]. For example, one of the applications of artificial intelligence (AI) and machine learning (ML) is that machine learning can comprise self-operating communication systems [35]. The application of this technology has generated important network paradigms, such as heterogeneous networks (HetNets) [36], ultra-dense networks, radio access networks (RAN) and technologies, such as terahertz (THz) in wireless networks and reconfigurable intelligent surfaces (RIS) [37] for spectrum management [38], user associations, routing optimizations, channel estimation, equalization and energy efficient network management and security [39]. Another application for AI in mobile networks is the design of hierarchical incentive mechanisms for federated machine learning. This application is mainly implemented using an IoT approach designed for the mobile applications of crowd detection in relation to reception and connectivity that IoT devices require [40]. Thus, the recent advances that have been achieved since the implementation of AI and ML in different communication systems have opened the doors and created a way for these two technologies to become fundamental parts of the design of 5G and 6G mobile network systems [41,42] as well as integrating MIMO into signal processing; thus, these processing capabilities are improved by facilitating additional functions required for the demand in mobile networks [43].

However, deep learning (DL) is a technique for implementing ML through the use of multilayer artificial neural networks (ANNs) [44]. This technique has allowed research that facilitates the use of DL as a deep learning technique that enables the capturing of network traffic data and thus allows the intelligent control of network traffic [45,46] by applying different techniques of deep learning, such as variations in neural network algorithms or LSTM (long short-term memory) [47]. Deep learning applications in mobile networks are broad; these applications range from monitoring, quality improvement, management and design, among others. One of the applications that has been studied recently is the use of semi-supervised machine learning to detect anomalies in cellular networks [48]. AI-based computer vision has also been studied using DL in 6G wireless networks, which use DL algorithms to solve different problems when recognizing images and objects via AI in 6G networks [49]. Thus, the use of DL in mobile networks has been considered a possible

solution for the high volume of data, traffic management and troubleshooting in mobile networks [13].

However, the term IoT refers to new proposals that allow connection and that can respond to the needs of IoT devices [30]. The other two terms corresponding to wireless communication refer to the type of communication, that is, wireless communication and optimization that corresponds to the built codes of machine learning used to improve the availability of the network, given that the majority of this form of technology and most IoT devices demand high performance and low latency [50]. The term IoT refers to the possibility of connecting different objects, such as equipment with sensors and cards for processing, storage and network capacity, which allows them to be connected to the network [51].

In the second quadrant are the less frequent but more current terms, thus indicating that they emerged in recent years. These terms include Beyond 5G, which suggests that they are networks that go beyond the characteristics of 5G networks; that is, these networks involve more than just communication and data transmission [52]. With 224 citations, one of the most-cited works with the term Beyond 5G (B5G) among its keywords is the research proposed by [53], who presents a new 3D cellular architecture specifically for unmanned aerial vehicles (UAVs) or drones; this architecture is based on a truncated octahedron structure for the cells, the spatial distribution of the equipment and minimizing latency, which resulted in the reuse of the frequency, thus improving the efficiency in the use of drones that are able to connect through wireless mobile networks. The authors of [23] analyze the different technologies and methodologies proposed in the literature examining the integration of nonorthogonal multiple access schemes via power domain [54]; this integration allows the connection of multiple users in the same block of resources by multiplexing either frequency or time, by facilitating communication technologies, such as machine learning, and it is expected that they respond to the demands of B5G networks [55].

Another term found in this quadrant is edge computing, which concerns the design of a network in such a way that the central network resources can be transferred to the perimeter in order to obtain a greater capacity of computing resources, allowing, in this case, the better management of management of IoT (Internet of Things) devices that may require greater capacity and availability of resources [23]. With this type of computing, it is possible to reduce latency by allowing the downloading and execution of certain tasks either locally or remotely, depending on the availability of the network or resources [56]. In terms of the emerging LSTM systems, there has also been work on identifying the best way to optimize the short-term memory of devices to improve the quality of service [57]. The next term is federated learning (FL), which focuses on a machine learning technique which trains models to allow collaboration in a decentralized way in order to build models that maintain data privacy [58]. FL algorithms allow devices to learn a model according to the data provided by an edge server [51]. The devices connected to this server can be found in the same geographical area or distributed in different areas, and the FL model takes the local data provided and thus provides feedback to the devices in the different areas [59].

Continuing with quadrant three, which contains the less frequent and less current terms, there are terms such as "neural networks", where neural networks are based on biology and allow machines to learn from observing data [57]. Neural networks have a great capacity to learn and solve different situations in signal processing and wireless communication [59]. These consist of an input layer and one or more hidden layers and an output layer, where each layer can contain one or more neurons [57]. These neurons consist of an activation function and several links that connect them with other neurons in different layers [59]. This network tries to find the optimal weight dataset that minimizes the error between the hypothesis function and the labels of the given dataset through forward and backward propagation. This technique can be used for constructing algorithms that enable the efficient management of networks, mainly regarding traffic [60]. For example, one of the techniques that has been used in deep learning algorithms is that of recurrent neural networks (RNNs) for the dynamic distribution of users with respect to traffic and

spatially distributing the connection of users from the data obtained by the call log [61]. Another technique, for example, is the feedforward neural network (FFNN), which can be implemented via the use of indicators for managing of the traffic of 5G networks, such as queries for success rate, propagation delay, overall discarded packets, power consumption, bandwidth usage, latency rate and network performance [62]. In this same quadrant, there is the term "energy efficiency", which refers to the efficient energy use of cellular networks, for example, turning off some of the carrier bands when network traffic is minimal [63].

Another term is "Mmwave", which refers to millimeter wave communications (mmWave), which is one of the main characteristics of 5G networks and refers to the way in which the beams are positioned allows a better capacity in the gain and thus balance the high propagation and compensate for the loss through the penetration of the millimeter wave bands, minimizing the time for transmission and energy consumption [64]. The term "deep reinforcement learning" is also found in this quadrant, which is a dynamic reservation technique and deep reinforcement learning that, in the case of mobile networks, allows the autonomous management of network resources, oriented according to the requirements of the same, based, for example, on the type of application that demands the availability of resources [65]. The next term is "big data", which is a technology that has been transformative in the recent developments and research of the latest generations of mobile networks, given that the technologies and techniques that have been recently implemented, developed and studied depend on the data that are being shared and travel through the network, which translates into all the traffic that travels through them [66]. It has also been used to capture network traffic data and use it for marketing purposes for the operators of different cellular networks [67]. The data processed from the use of big data lead to the next term, "resource allocation", which is about looking for the best storage management. For example, proactive caching is concerned with the storage of data from mobile network traffic from which only specific data can be selected, which provides information on connectivity and user resource demand, thus optimizing storage space [68].

In the last quadrant, which comprises the most frequent but least current terms, the term "reinforcement learning" can be found, which is derived from one of the deep learning techniques already explained as well as network slicing. Reinforcement learning is usually used in communication networks for authenticating and validating devices in the network. These learning algorithms are either parametric, nonparametric, supervised, unsupervised or involve reinforcement learning to facilitate intelligent authentication and make the connection of devices in the network more reliable and profitable [68]. Additionally, reinforcement learning has been applied to the intelligent allocation of resources in networks [69]. Combined with network slicing, algorithms are used alongside the ability to automatically learn patterns in the connection requests of the users in such a way that the routes that are frequently requested can be detected and thus users ca be intelligently distributed along the connection routes and thus greater efficiency in allocating resources in the network can be achieved [70].

However, the main application that has been provided in research on reinforcement learning is assisted caching for heterogeneous mobile networks [71]. The term "network slicing", which lies between quadrants three and four, refers to network segmentation to address different needs in administrating and orchestrating network services and applications [72]. Based on the above, network slicing can be understood as a technology that allows the wide demand of services in mobile networks running on SDNs in a shared network infrastructure [73]. In this way, one of the main approaches of network slicing is identifying the applications to perform a specific segmentation of the network according to the connected application [74]. This approach is supported by reinforcement learning.

As can be observed in the relationship between the key terms and their recurrence in research for mobile networks and the use of machine learning techniques, most of them focus on the optimization of resources and traffic to guarantee the best management and availability of the network due to the high demand for resources and greater amount of traffic generated by the many IoT devices that are being developed on the market. Figure 8

shows how the aforementioned terms interact, that is, a network of key terms used the different studies has been developed. The main term and central nexus of the network is that of "fifth generation mobile communications" or 5G networks; however, there is already specific talk about 6G networks because 4G networks are currently still in transition. LTE is still working on the automation of 5G networks [75]. Thus, automatic learning algorithms have been implemented to reduce latency in the network, and these algorithms for 5G mainly use different deep learning techniques for network optimization, as seen in the networks formed [76].

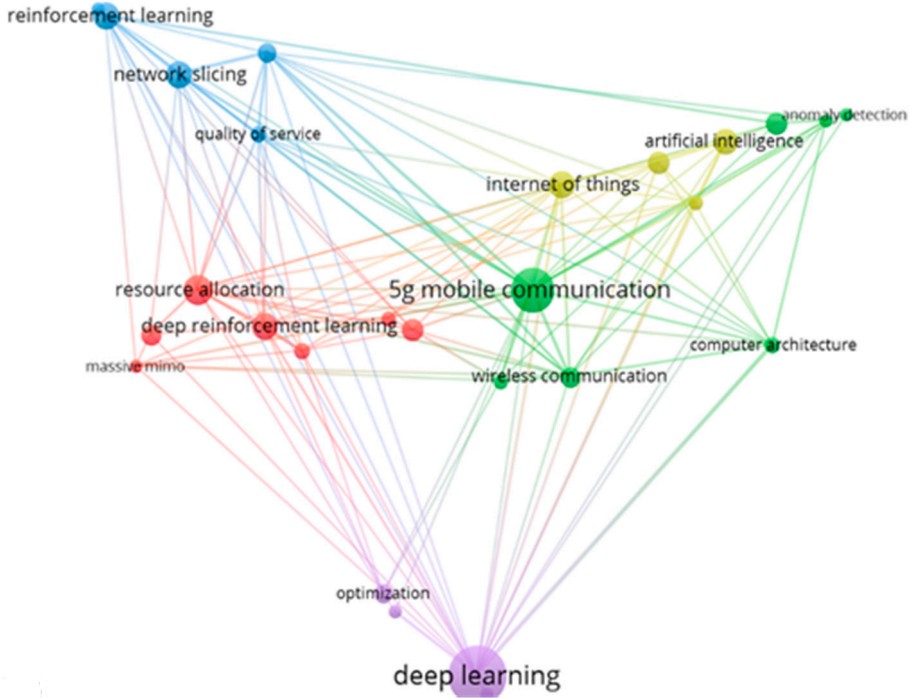

**Figure 8.** Network of keywords on mobile networks and machine learning. Own elaboration.

The above network of keywords is concerned with improving the quality of service provision of mobile networks from segmentation through network slicing by using deep reinforcement learning (DRL) algorithms for space allocation and division in the network, according to the consumption of resources or the needs of the users in the network [69]. Then, there is a network consisting mainly of the terms "deep reinforcement learning", "resource allocation" and "massive MIMO". These terms are connected due to recent research focused on the use of deep learning reinforcement algorithms to improve the allocation of resources in the network and thus guarantee a better special distribution of users; this study is supported by one of the techniques used to modulate the signal and thus appropriately assign the characteristics to the signal according to the needs [77]. Additionally, as revealed in the quadrants of the previous figure, the use of artificial intelligence technology in the latest-generation networks is aimed mainly at guaranteeing the adequate availability of resources due to the massive number of IoT devices that demand considerable resources [78]. Finally, the terms "anomalies", "computer architecture" and "wireless communications", which focus on research that proposes a framework of solutions for access to radio communications [79].

As for the most used keywords per year, as Figure 9 reveals, starting with the year 2006, the most used keyword was "UMTS", which refers to the third generation (3G) of mobile networks, specifically to the universal mobile telecommunications system, which was one of the technologies most used by mobile devices after the year 2000 approximately [80]. At that time, studies were carried out through fuzzy learning for the better management of the mobile network [81]. Then, in 2012, the most commonly used keyword was "SPIM", which

translates as SPAM detection on instant messaging, and the studies carried out focused on how to improve and characterize this type of message to mitigate the overload of the server [82]. In 2014, the term "web browsing" was the most used keyword, focusing efforts on research to model the quality of the browsing experience on the cellular network [83].

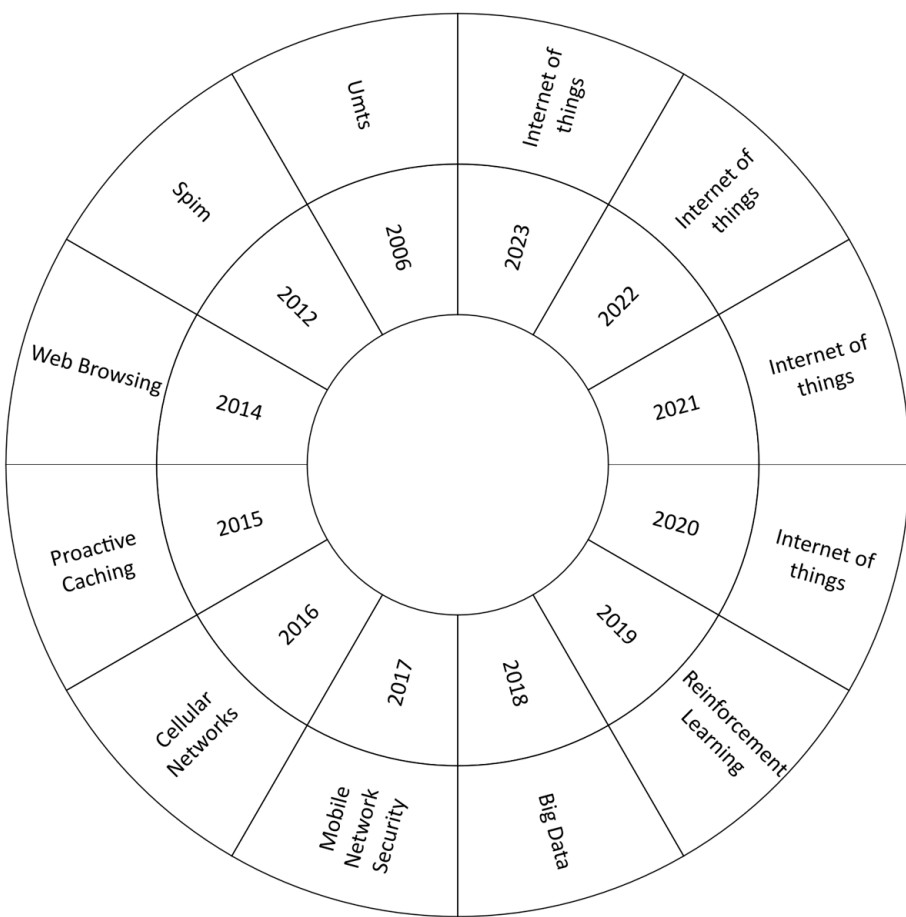

**Figure 9.** Keywords by year on the topic of mobile networks and machine learning. Own image.

For 2015, one of the terms used was "proactive caching", and the related research carried out focused on the use of big data to improve the proactive perspective of caching to improve the hosting of resources [84]. In 2016, "cellular networks" emerged as the most commonly used key term, where studies focused mainly on finding solutions for network traffic management [63] and methods for dynamic network configuration [85]. However, some studies also emerged that were already beginning to discuss 5G networks [86]. Based on the research carried out in 2016, from 2017 to 2022, "5G" (for mobile networks) emerged as the most used keyword, as scholars began to talk about the virtualization of networks for their more autonomous management [87] and radio-cognitive 5G networks [88]. In 2019, research on 5G began to focus on the use of deep learning algorithms for applications that allowed better quality, mainly a reduction in the amount of latency in the network [89]. In 2020, studies continued to focus on improving deep learning techniques to improve the use of resources, and in these studies, they began to talk about energy efficiency [90]. By 2021, 5G research had begun to discuss the next generation of computing [91] and how mobile networks should be prepared for it along with networks for the IoT and the great demand for network resources [92]. Finally, for the year 2023, in the studies that have been carried out thus far, the most used keyword has been "password security", involving talk of authentication systems for network security [93].

## 4. Discussion

The growing interest in the current subject led to increased publications during 2019 and 2020, and the increased publication of research on how to increase the availability of resources in instances where many devices are connected the same network [71] was possibly due to the pandemic caused by the new coronavirus disease 2019 (COVID-19). During this time, smart devices became an integral part of human life, thus requiring highly available and scalable networks, which demand high-speed and real-time response [3].

Regarding key terms, recent research has focused on generating algorithms for recognizing and learning data in the network [6] and trained models start from the available data in order to improve connections to the mobile network, thus improving quality and latency and allowing increasingly faster responses from the network [13]. However, one of the great advantages that this technology is a set of concepts for automated network management, not only to improve the quality of service but also to reduce network management burdens on network administrators [94].

It is essential to mention that the present literature review extracted information from the export processes provided by Scopus and Web of Science databases. The information was then stored, managed and organized using Microsoft Excel® version 2304. This approach ensured proper metadata management, as identified in the results section, as well as in the discussion and conclusion sections. It allowed for the recognition of key research trends, reflected in the prominent keywords throughout the chronological course of the investigative history on machine learning in mobile network connections, including emerging trends in the field. This, in turn, led to the establishment of the previously proposed research agenda (see Figure 10). Consequently, it presents a series of investigative challenges for future work to address the current conceptual gaps and expand knowledge in the field.

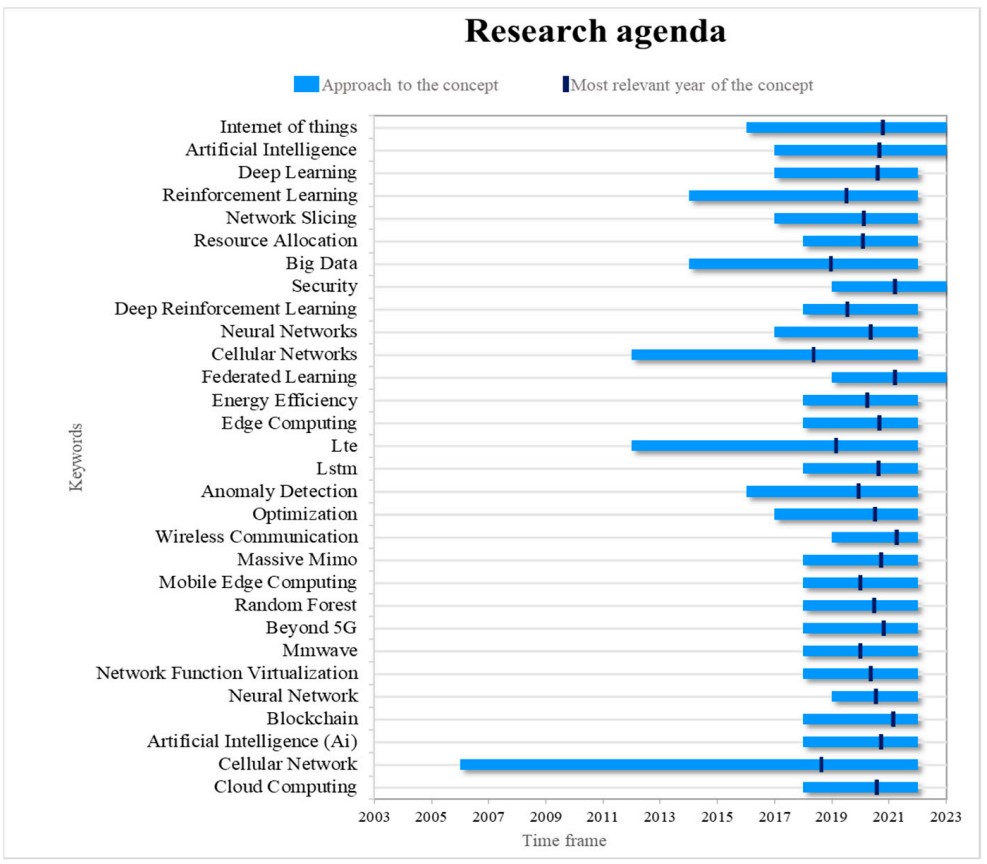

**Figure 10.** Research agenda on the topic of mobile networks and machine learning. Own image.

Regarding the research questions initially raised in this study, the first question corresponds to the year with the highest productivity in the research field. Although, since 2015, much scientific productivity has been recorded through publications in the area of machine learning and mobile networks, it was in the years 2019 and 2020 where the greatest productivity occurred. This suggests that as Industry 4.0 took on topics such as artificial intelligence and IoT, research associated with the response to the high technological demand began to emerge.

The following question refers to the main references in the scientific literature, i.e., the journals belonging to the IEEE, mainly IEEE Access, are those with the largest number of articles published in the area of machine learning and mobile networks. This is possibly because one of the main areas of knowledge of this magazine is electronic engineering and telecommunications. Regarding the authors, it was Benmis, M., and Debbah, M., who have published the most on the subject and have not only the largest number of publications but also citations.

Regarding the conceptual evolution, starting from 2003, when the universal mobile telecommunications system (UMTS) was still being discussed, along with cellular networks, until 2017, when research began to focus on the security protocols of mobile networks. Subsequently, in 2018, big data was integrated into data analysis through mobile network traffic. In 2019, the application of techniques that allow the optimization of networks, specifically the reinforced learning technique, becomes more relevant. As of 2020, the application of different techniques, tools and mechanisms that allow the stability, security and quality of networks along with the emergence of the Internet of Things and the possible increase in traffic in mobile networks have taken center stage.

The main emerging and growing topics include the Internet of Things, artificial intelligence and the different deep learning techniques applied to the design of mobile networks. Thus, an investigative agenda arises from the application of techniques, such as neural networks and federated learning, and is applied mainly to factors such as network security, Beyond 5G and edge computing.

In addition to the contribution by providing the main research trends in the field of machine learning and mobile networks, this paper provides key elements for the development of future research by creating keyword clusters and a research agenda based on key terms used in previous investigations. Although machine learning and artificial intelligence have been widely addressed by different studies, for future frameworks, the results of this investigation will allow those interested in the subject to know which subject to focus on and to understand what applications to work on, as well as knowing which are the most relevant contributions that have been made on the topic. With the above, this work is useful as a referential framework for future research and the development of degree projects for future professionals who are being trained in wireless communications. Finally, Table 2 shows the 10 articles with the highest number of citations in Scopus. Most of these are review articles where the main techniques and applications of machine learning for wireless networks are discussed.

**Table 2.** Main techniques and applications discussed in the articles with the greatest impact.

| N | Citation | Main Contribution | Limitations | Methodology | Citation Number | Technique | Technique Approach | Application |
|---|---|---|---|---|---|---|---|---|
| 1 | [13] | Based on an exhaustive literature review, the authors provide different options to adapt deep learning models to mobile device networks in general and highlight the different problems to be solved, thereby opening up in-depth research in the field of knowledge of mobile networks and machine learning. | Although the work focuses on deep learning, other machine learning techniques used in wireless networks could be compared. | Literature Review | 825 | Deep Learning | Deep Learning-Driven Network-Level Mobile Data Analysis; Deep Learning-Driven App-Level Mobile Data Analysis; Deep Learning-Driven User Mobility Analysis; Deep Learning Driven User Localization; Deep Learning-Driven Wireless Sensor Networks; Deep Learning-Driven Network Control; Deep Learning-Driven Network Security; Deep Learning-Driven Signal Processing; Emerging Deep Learning-Driven Mobile Network Application | Wireless Networks |
| 2 | [19] | The authors present a detailed review of DRL approaches proposed to address emerging problems in communication networks, such as dynamic network access, data rate control, wireless caching, data offloading, network security and connectivity preservation. Additionally, the authors present DRL applications for traffic routing, resource sharing and data collection. | Although the work focuses on deep reinforcement learning, other machine learning techniques used in wireless networks could be compared. | Literature Review | 806 | Deep Reinforcement Learning | Deep Deterministic Policy Gradient Q-Learning for Continuous Action; Deep Recurrent Q-Learning for POMDPs; Deep SARSA Learning; Deep Q-Learning for Markov Games | Communications Network |
| 3 | [14] | The authors present five lines of future research related to massive MIMO, digital beamforming and/or antenna arrays. These five lines focus on proposals for extremely large aperture arrays, holographic massive MIMO, six-dimensional positioning, large-scale MIMO radar and intelligent massive MIMO. | Although the authors provide windows for future research around antenna arrays and massive MIMO. The authors do not consider multiple options regarding antenna array and do not compare MIMO with other techniques; although they talk about the next generation of communications, they do not consider a wide field on 6G communications and dedicate only a small space to it. | Literature Review | 362 | Machine Learning | Reinforcement Learning | Antenna Arrays |
| 4 | [15] | In this review article, the authors describe up-to-date research on the integration of multi-access edge computing with new technologies to be deployed in 5G. | While the authors provide a complete framework on multi-access edge computing features, they focus on applications, needs and features, leaving less room for machine learning techniques applied in current research advances. | Literature Review | 354 | Machine Learning | Unsupervised Learning; Supervised Learning; Reinforcement Learning | 5G Network |
| 5 | [17] | The authors present an overview of the sixth generation (6G) system based on the following possibilities: usage scenarios, requirements, key performance indicators (KPIs), architecture and enabling technologies, based on the projection of mobile traffic to 2030. | While the authors provide a complete framework on the features and a strategic path for 6G, starting from the possible applications, use cases and scenarios, a reduced part is left for the possible machine learning techniques and comparison of the same in 6G. | Literature Review | 323 | Artificial Intelligence | Block Chain; Digital Twins; Intelligent Edge Computing; | 6G Network |
| 6 | [20] | Particular challenges present and future research needed in control systems, networks and computing, as well as for the adoption of machine learning in an I-IoT context | This article focuses on the characteristics of the architecture necessary for IoT, taking into account the possible high traffic demand that will be required to facilitate the connection of these devices. However, it does not focus on machine learning techniques that may allow the best management of networks for the connection of IoT devices. | Literature Review | 315 | Machine Learning | Unsupervised Learning; Supervised Learning; Reinforcement Learning | IoT |
| 7 | [95] | In this research, the authors carry out a general description of the most common machine learning techniques applied to cellular networks, classifying the ML solution applied according to the usage. | Different machine learning techniques are discussed; however, only a final reference to deep learning is made, without expanding the possibilities of the application of deep learning techniques. | Literature Review | 299 | Machine Learning | Supervised Learning (k-Nearest Neighbor; Neural Networks; Bayes' Theory; Support Vector Machine; Decision Trees); Unsupervised Learning (Anomaly Detectors; Self Organizing Maps; K-Means); Reinforcement Learning | Cellular Networks |
| 8 | [30] | The authors indicate different potentials of cloud-based machine learning (ML) for the architectural deployment of 5G by presenting different case studies and applications that demonstrate the potential of edge ML in 5G. | The authors focus their research on edge ML, so the study focuses on providing an overview of future research on edge ML without considering other types of architectures for wireless networks, although they reflect on different types of wireless networks. | Literature Review | 269 | Machine Learning | Supervised Learning (k-Nearest Neighbor; Neural Networks; Bayes' Theory; Support Vector Machine; Decision Trees); Unsupervised Learning (Anomaly Detectors; Self Organizing Maps; K-Means); Reinforcement Learning | Wireless Network |
| 9 | [96] | The authors provide a description of the possible enablers of 6G networks from the domain of theoretical elements of machine learning (ML), quantum computing (QC) and quantum ML (QML). The authors propose possible challenges for 6G networks, benefits and usages for applications in Beyond 5G networks. | This is a work more focused on the future of communication networks based on the application of quantum computing techniques; therefore, less emphasis is placed on recent use cases of machine learning applied to mobile networks. | Literature Review | 269 | Quantum Machine Learning | Supervised Learning; Semi-supervised and Unsupervised Learning; Reinforcement Learning; Genetic programming; Learning Requirements and Capability; Deep Neural Networks; Deep Transfer Learning; Deep Unfolding; Deep Learning for Cognitive Communications | Beyond 5G |
| 10 | [53] | The authors propose a 3D cellular architecture for drone base station network planning and minimum latency cell association for user equipment drones, through a manageable method based on the notion of truncated octahedral shapes, allowing for the complete coverage for a given space with a minimum number of drone base stations. | The research is based solely on a proposal for drones that can be replicated for unmanned aerial vehicles; however, it does not consider other mobile equipment. | Kernel density estimation | 266 | Machine Learning | | 3D Wireless Cellular Network |

### 5. Conclusions

Regarding the number of publications per year, the increased interest in the subject between 2019 and 2020 was due to the pandemic caused by new coronavirus disease 2019 (COVID-19). Regarding the publications by journal, the *IEEE Access*, being focused on engineering, is the journal predominantly publishing papers whose area of knowledge is entirely focused on engineering. Regarding the main countries, the United States and China, which compete for the coverage of 5G networks, are the ones that are responsible for highest level of manufacturing of devices for mobile networks; thus, it is understood why the research in this area is mainly from these countries.

The progress in the study of mobile networks has been developed as the demand for network resources has increased, which is in response to the large number of IoT devices necessary for the new infrastructure of cities, which is not only intelligent but responsible in terms of the environment. This is how the change in the different types of modulations that have arisen is observed, where it is not only necessary to modulate by frequency and space, moving away from the structure of mobile communications via cells, but also resulting in a communication structure that responds to data traffic and learns according to its flow. Therefore, it can be observed how research on the subject has moved from focusing on the behavior of the signal and the type of antennas needed to focusing on the use of programming codes that allow the best optimization and automation of network resources.

One of the main aspects in recent research is that most have focused on the use of different AI techniques for improvement in resource management and network availability. However, although there is talk of the use of MIMO, for example, for signal processing, little is discussed with respect to the direction and range of the antennas that will be available. However, it would be interesting to focus efforts not only on the development of the optimization and automation techniques for resource management but also on a better directivity and reach in the signal. This could represent one of the greatest challenges in the area of knowledge, since there is constant discussion in relation to the gap that exists between software and hardware advancement in the area of telecommunications.

Consequently, it would be interesting to advance in the proposal of different antenna arrays that allow better directivity and range in high traffic conditions and not only concentrate on the optimization of resources and network availability. Currently, miniaturized antenna designs called micro-ribbon antennas or patch antennas have been proposed through arrays from materials such as FR4 and graphene, which are not only low power but also efficient for use in IoT devices due to their size. This type of antenna would not only be suitable for data transmission in a network, but they would also allow a sustainable IT infrastructure due to the low cost of their design and low energy use. For this type of antenna to be used in next-generation mobile networks, it is still necessary to advance research further; however, there are already advances in scientific papers in which the use of different algorithms of deep learning is demonstrated for the design of this type of antenna in other applications, so it would be possible to improve their directivity and range conditions. New machine learning techniques in patch antenna arrays has not been widely discussed but is a topic that could be covered in the future research in the field.

On the other hand, the 5G mobile networks and the emergence of 6G networks are based on the optimization of algorithms that will allow not only connectivity but also improvement in quality, service availability and speed improvement, in addition to self-management. These characteristics are the focus of future research due to the increasingly accelerated growth of the IoT and its demand for increasingly more bandwidth. For future research on identifying trends in machine learning and mobile networks, it is possible to carry out a systematic literature review, in which criteria can be established that enable the selection of specific articles that, in turn, enable the selection of variables, materials and specific techniques within the subject.

**Author Contributions:** Conceptualization, V.G.-P. and A.V.-A.; methodology, A.V.-A. and V.G.-P.; software, A.V.-A.; validation, V.G.-P., J.C.P.-V. and D.A.-B.; formal analysis, J.J.F.C.; investigation, P.A.R.-C.; resources, A.M.R.C.; data curation, D.A.-B.; writing—original draft preparation, V.G.-P.; writing—review and editing, A.V.-A.; visualization, J.C.P.-V.; supervision, V.G.-P.; project administration, A.V.-A.; funding acquisition, J.J.F.C. All authors have read and agreed to the published version of the manuscript.

**Funding:** This research was funded by the Corporación Universitaria Americana (Colombia) and the Universidad Señor de Sipán (Perú). The APC was funded by Universidad Señor de Sipán (Perú).

**Informed Consent Statement:** Not applicable.

**Data Availability Statement:** The data may be provided free of charge to interested readers by requesting the correspondence author's email.

**Conflicts of Interest:** The authors declare no conflict of interest.

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
