# Peer review of "Research Trends in the Use of Machine Learning Applied in Mobile Networks: A Bibliometric Approach and Research Agenda"

_informatics, doi:10.3390/informatics10030073_

Round 1

Reviewer 1 Report

The work presents a Systematic Literature Review of the application of machine learning to wireless mobile networks. In general, the review posed a significant impact and it is a good reference material in the domain. However, there are a few complex sentences that make the reading of the work a little difficult. The authors should revise such long sentences and improve the coherence and general flow of the paper. Here are a few suggestions to improve the quality of the paper:

1.     Can the topic be modified? Especially the part that says, “Use of Machine Learning Applied in the Connection of Mobile Networks”. The topic may be rephrased

2.     What is the main novelty and contribution of this review paper? It is a well-known fact that machine learning algorithms have been widely advantageous and useful in the design of wireless mobile networks. Apart from examining the research trends, what are the other contributions?

3.     The authors should improve on the literature review section of this manuscript. In section I, a table should be developed which compares the contributions of this work to other recent manuscripts in this field especially review papers where the use of machine learning in wireless mobile networks has been described and discussed. The focus and coverage of the work, its limitations should also be included in the table.

4.     This work lacks methodology. I cannot find the design framework. The methodology is missing, even as a review paper, there is need to include the design framework in a greater detail than what is currently given in the manuscript.

5.     Four research questions were introduced but the conclusion was not drawn based on these research questions. Was this intentional? The authors should please draw conclusions based on the research questions introduced in the study.

6.     In the conclusion section, a brief summary of the key findings from the survey is requested. It is imperative to state the key takeaways from the extensive survey and their implications for further research. In my opinion, the current conclusion is limited. What do you think?

7.     So many abbreviations, the author should create a list of abbreviations of all abbreviated terms. This will help the flow and the readability of the manuscript.

8.     Can the author examine the research trends based on the use of several machine learning algorithms in wireless mobile networks? Can the authors put this in a table or a chart?

Author Response

May 18, 2023

Dear

Design – Editorial Team

Kind regards

In accordance with the suggestions of the reviewers in our article “Use of Machine Learning Applied in the Connection of Mobile Networks: A Bibliometric Approach and Research Agenda, the following changes were made, properly marked with red letters in the article:

Comment

Response

1. Can the topic be modified? Especially the part that says, “Use of Machine Learning Applied in the Connection of Mobile Networks”. The topic may be rephrased

The title is changed to: Research Trends In The Use of Machine Learning Applied in Mobile Networks: A Bibliometric Approach and Research Agenda

2. What is the main novelty and contribution of this review paper? It is a well-known fact that machine learning algorithms have been widely advantageous and useful in the design of wireless mobile networks. Apart from examining the research trends, what are the other contributions?

A section is added at the end of the conclusions indicating the contribution of the work developed

3. The authors should improve on the literature review section of this manuscript. In section I, a table should be developed which compares the contributions of this work to other recent manuscripts in this field especially review papers where the use of machine learning in wireless mobile networks has been described and discussed. The focus and coverage of the work, its limitations should also be included in the table.

Table 1 is added in the discussion section, where the most cited articles are described, finding that the majority correspond to literature reviews. The techniques and applications presented in them are presented.

4. This work lacks methodology. I cannot find the design framework. The methodology is missing, even as a review paper, there is need to include the design framework in a greater detail than what is currently given in the manuscript.

The article contains the methodology section mentioned by the reviewer, specifically titled "2. Material and Methods", setting out the parameters suggested by the PRISMA declaration, as mentioned, for which reason they are detailed: eligibility criteria, sources of information , search strategy, data management and selection process.

5. Four research questions were introduced but the conclusion was not drawn based on these research questions. Was this intentional? The authors should please draw conclusions based on the research questions introduced in the study.

The questions are answered in the conclusion section.

6. In the conclusion section, a brief summary of the key findings from the survey is requested. It is imperative to state the key takeaways from the extensive survey and their implications for further research. In my opinion, the current conclusion is limited. What do you think?

A paragraph is added in the conclusions section, where it is explained that, more than a form or survey, the default export format of both databases is used, through which the information is extracted and managed, so that could raise the research agenda and, therefore, propose a series of research challenges for future work

7. So many abbreviations, the author should create a list of abbreviations of all abbreviated terms. This will help the flow and the readability of the manuscript.

The table of abbreviations is added at the end of the introduction.

8. Can the author examine the research trends based on the use of several machine learning algorithms in wireless mobile networks? Can the authors put this in a table or a chart?

The main techniques and applications are added in Table 1 in the final part of the discussion and conclusions.

We look forward to your comments and hope to hear from you soon.

Thank you very much

_

The authors

Reviewer 2 Report

The authors offer a commendable review of the use of machine learning in connectivity in mobile networks.

The selection and review criteria are sound, and the offered analysis is interesting -- especially from a research management perspective.

As provided, the work merits publication. However, it is elemental that the authors perform a careful proofreading of the writing, e.g., in the discussion and conclusion section, the choice of the journal, "the IEEE", and the word "incosequence" indicate a hasty write-up.

Author Response

May 18, 2023

Dear

Design – Editorial Team

Kind regards

In accordance with the suggestions of the reviewers in our article “Use of Machine Learning Applied in the Connection of Mobile Networks: A Bibliometric Approach and Research Agenda”, the following changes were made, properly marked with red letters in the article:

Comment

Response

As provided, the work merits publication. However, it is elemental that the authors perform a careful proofreading of the writing, e.g., in the discussion and conclusion section, the choice of the journal, "the IEEE", and the word "incosequence"; indicate a hasty write-up.

Editing is done

We look forward to your comments and hope to hear from you soon.

Thank you very much

_

The authors

Reviewer 3 Report

The article examines the research trends in the development of mobile networks from machine learning. The methodological approach starts from an analysis of 260 academic documents selected from the Scopus and Web of Science databases and is based on the parameters of the Preferred Reporting Items for Systematic Reviews and Meta-Analyses (PRISMA) statement. Moreover, the article claims it provides a research agenda for future work on mobile networks and machine learning.

While a number of reviews have been written on the topic, the systematic review approach is fresh and interesting. However, unfortunately, the manuscript falls short on its promises. The publication year, journal, author, and keyword analysis, while nice to read, don't contribute much new to the ongoing discussion on the use of machine learning in mobile networks. Moreover, the manuscript provides shallow and occasionally misleading conclusions on the articles it reviews.

For example, the article refers to edge computing a way of allowing "connection and remote management of IoT devices", missing the point that edge/fog means the computational capacity available in the network infrastructure. Moreover, according to the manuscript, "emerging" LSTM involves "work on finding the best way to optimise the short-term memory of devices". There's no mention of LSTM being a neural network architecture, introduced years ago and now largely surpassed by transformers.  

In the same vein, federated learning, a distributed machine learning method, is referred to as a machine learning technique that "trains an algorithm to manage resources", allowing "devices to learn a model according to the data provided by and edge server". Neural networks, according to the manuscript, are "techniques used for constructing algorithms that enable efficient management of networks, mainly traffic". The manuscript is unfortunately full of such examples.

The research agenda, given in the final section, also falls short of providing novel ideas for future research, listing general topics such as IoT, AI, deep learning, reinforcement learning, and network slicing, reading just the top items on the list. In the same vein, the main conclusion of the article (as given in the introduction section) is that "the main trends in the use of Industry 4.0 technologies, such as machine learning, involve designing and developing algorithms that can learn from the data provided by network traffic and that allow the current and next-generation networks to self-manage". While true, the conclusion offers nothing new to the research community.

Author Response

May 18, 2023

Dear

Design – Editorial Team

Kind regards

In accordance with the suggestions of the reviewers in our article “Use of Machine Learning Applied in the Connection of Mobile Networks: A Bibliometric Approach and Research Agenda”, the following changes were made, properly marked with red letters in the article:

Comment

Response

While a number of reviews have been written on the topic, the systematic review approach is fresh and interesting. However, unfortunately, the manuscript falls short on its promises. The publication year, journal, author, and keyword analysis, while nice to read, don't contribute much new to the ongoing discussion on the use of machine learning in mobile networks. Moreover, the manuscript provides shallow and occasionally misleading conclusions on the articles it reviews. For example, the article refers to edge computing a way of allowing "connection and remote management of IoT devices", missing the point that edge/fog means the computational capacity available in the network infrastructure. Moreover, according to the manuscript, "emerging"; LSTM involves "work on finding the best way to optimise the short-term memory of devices". There's no mention of LSTM being a neural network architecture, introduced years ago and now largely surpassed by transformers. In the same vein, federated learning, a distributed machine learning method, is referred to as a machine learning technique that "trains an algorithm to manage resources", allowing "devices to learn a model according to the data provided by and edge server";. Neural networks, according to the manuscript, are "techniques used for constructing algorithms that enable efficient management of networks, mainly traffic". The manuscript is unfortunately full of such examples.

These examples were corrected in the text.

The research agenda, given in the final section, also falls short of providing novel ideas for future research, listing general topics such as IoT, AI, deep learning, reinforcement learning, and network slicing, reading just the top items on the list. In the same vein, the main conclusion of the article (as given in the introduction section) is that "the main trends in the use of Industry 4.0 technologies, such as machine learning, involve designing and developing algorithms that can learn from the data provided by network traffic and that allow the current and next-generation networks to self-manage". While true, the conclusion offers nothing new to the research community.

The conclusions section was expanded and a table with the main contributions to the research was added.

We look forward to your comments and hope to hear from you soon.

Thank you very much

_

The authors

Round 2

Reviewer 1 Report

The authors have satisfactorily made all the revisions as requested by the reviewer. Some general minor English correction is required.